# Microstructure and Tensile Properties of Melt-Spun Filaments of Polybutene-1 and Butene-1/Ethylene Copolymer

**DOI:** 10.3390/polym15183729

**Published:** 2023-09-11

**Authors:** Jianrong Li, Yongna Qiao, Hao Zhang, Yifei Zheng, Zheng Tang, Zhenye Zeng, Pingping Yao, Feng Bao, Huichao Liu, Jiali Yu, Caizhen Zhu, Jian Xu

**Affiliations:** 1Institute of Low-Dimensional Materials Genome Initiative, College of Chemistry and Environmental Engineering, Shenzhen University, Shenzhen 518060, China; chemljr@126.com (J.L.); yongna_qiao@sina.com (Y.Q.); zhanghao5@sinap.ac.cn (H.Z.); xiaofeigeonline@163.com (Y.Z.); tangzheng1104@163.com (Z.T.); zengzhye@163.com (Z.Z.); ppyao@szu.edu.cn (P.Y.); bfisvip@163.com (F.B.); huichaoliu@szu.edu.cn (H.L.); jlyyhx@163.com (J.Y.); czzhu@szu.edu.cn (C.Z.); 2College of Textile Science and Engineering (International Institute of Silk), Zhejiang Sci-Tech University, Hangzhou 310018, China

**Keywords:** polybutene-1 filaments, ethylene co-units, mechanical properties, SAXS/WAXD

## Abstract

Polybutene-1 with form I crystals exhibits excellent creep resistance and environmental stress crack resistance. The filaments of polybutene-1 and its random copolymer with 4 mol% ethylene co-units were produced via extrusion melt spinning, which are expected to be in form I states and show outstanding mechanical properties. The variances in microstructure, crystallization–melting behavior, and mechanical properties between homopolymer and copolymer filaments were analyzed using SEM, SAXS/WAXD, DSC, and tensile tests. The crystallization of form II and subsequent phase transition into form I finished after the melt-spinning process in the copolymer sample while small amounts of form II crystals remained in homopolymer filaments. Surprisingly, copolymer filaments exhibited higher tensile strength and Young’s modulus than homopolymer filaments, while the homopolymer films showed better mechanical properties than copolymer films. The high degree of orientation and long fibrous crystals play a critical role in the superior properties of copolymer filaments. The results indicate that the existence of ethylene increases the chain flexibility and benefits the formation of intercrystalline links during spinning, which contributes to an enhancement of mechanical properties. The structure–property correlation of melt-spun PB-1 filaments provides a reference for the development of polymer fibers with excellent creep resistance.

## 1. Introduction

High-performance fibers and their textiles, known for their high specific strength, high modulus, and corrosion resistance, are extensively utilized in various industries such as aerospace, automobile manufacturing, wind power generation, individual protection, and civil engineering construction [1,2,3]. These materials play a crucial role in the modernization of national defense and the development of the national economy. The objective of polymer processing into fibers is to optimize the microstructure or morphology of the polymer material in order to fulfill the precise mechanical properties required for diverse applications. Hearle and Greer [4] identified crucial factors that significantly influence the ultimate mechanical characteristics of fibers. These factors encompass the degree of order, the extent to which ordered phase domains are localized, the aspect ratio and size of localized units, the degree of orientation, and the molecular chain extensibility. In a general sense, the spinning process, commonly referred to as tensile flow, can be employed to accomplish the aforementioned objectives for flexible linear polymers [5]. This procedure entails the extension of the polymer chain in order to generate elongated fiber crystals. Polybutene-1 (PB-1), a flexible linear polymer, exhibits exceptional mechanical properties, remarkable creep resistance, and environmental stress crack resistance, among other outstanding characteristics, when compared with other Polyolefins. Consequently, extensive applications of PB-1 have been found in various fields, including geothermal pipes, easy-peel films, and food packaging, earning the reputation of “plastic gold” for PB-1 due to its excellent performance and high price [6]. PB-1 exhibited four different crystalline modifications: I, II, III, and I′ [6,7,8,9,10,11]. Previous studies have confirmed that melt processing of PB-1 mainly generates form II with an 11/3 helical conformation [12] and form I with a 3/1 helical conformation [13]; thus, the two crystal forms are of practical significance. During the processing of PB-1, the metastable form II is predominantly formed, exhibiting inferior mechanical properties. After a few weeks, form II undergoes a spontaneous and irreversible transformation into the thermodynamically stable form I with enhanced mechanical properties at ambient temperature [13,14]. The two-step phase transition “melt-form II-form I” results in a high production cost of PB-1. Due to the distinctive polymorphism behavior of PB-1, a significant portion of the literature was dedicated to employing hot-pressing film in order to explore methodologies for expediting crystal transformation. Hence, there is currently limited research on PB-1 spinning even though PB-1 filaments in form I with outstanding creep resistance can be expected to be obtained after the spinning process.

To enhance the performance of PB-1 material, researchers aim to accelerate the process of obtaining crystal form I and to increase its content by modifying external conditions such as external or thermal stress [15,16,17,18,19,20], high pressure [21], pressured CO_2_ [22,23,24], introduction of ionic functional groups [25], and copolymerization with random 1-alkene co-units with less than five carbon atoms [20,26,27]. According to Cavallo et al. [18], during the crystal transformation induced by external or thermal stress, stress controls the initial stage of transformation. The form II lamellae perpendicular to the tensile direction transform first. Once a critical point is reached, the stress is borne by the transformed form I and the stress dependence of crystal transformation disappears. In the later stage, crystal transformation is closely related to the plastic deformation process. Stress promotes the nucleation of crystal form I, and the lamellae of crystal form II undergo shear deformation, fracture, and transform into fibrous crystals [28]. In the case of PB-1, the melt-spinning process can be employed to attain the required stretching flow for the purpose of aligning the flexible chains and creating extended fiber crystals.

Melt spinning, as a physical method, has less impact on the environment, relative simplicity, and high speed, and is the most widely used industrial process for fiber formation [29]. The melt-spinning process involves applying drawing force to the polybutene-1 melt and crystallization through stretching. Polymer molecules and crystals align parallel to the fiber axis, providing unique properties to the final filaments. Choi and White [30] compared the development of crystallization and crystalline orientation in the melt spinning of four polyolefins, including PE, PP, PB-1, and P4MP, which likewise noted the crystalline transition behavior of PB-1 and showed that the birefringence of PB-1 fiber was lower in comparison to that of PE fiber. The study initially explored the difference between the orientation of PB-1 fiber and the well-known PE fiber but did not explore the relationship between microstructure and properties. Samon et al. [31,32] examined the structure of PB-1 fiber as they were being produced with spinning speeds up to 250 m/min, followed by annealing at room temperature. They found that higher spinning speeds caused an increase in crystallinity during the annealing step. Ortiz et al. [33] showed that melt-spun PB-1 fiber has high modulus and toughness compared with polypropylene or polyester fiber when being spun at similar speeds (250–2500 m/min) and the properties of polybutene-1 nonwoven mats prepared with them changed less on aging. The crystallization behavior of PB-1 fiber membrane was studied by Lee et al. [34] using electrospinning. Although electrospinning technology has been used to produce PB-1 fiber membranes, the microstructure of PB-1 fibers—specifically, the long period of lamellae or the degree of orientation of PB-1 fiber—has not been extensively explored due to the limitations of the technology. The majority of the above works were conducted on the crystallization behavior of homopolymer fibers. Furthermore, copolymerization has been utilized as a means to decrease the base cost and enhance the processability of polymers. For example, polyethylene glycol side chains provide an option for adjusting phase transition without sacrificing the crystallization performance of butene-1 copolymer [35]. The properties of copolymers are predominantly influenced by the inherent properties of the components that are copolymerized. Adding five carbon-ring-like units to the main chain of polybutene-1 results in a melt memory effect and different polymorph selection behavior during melt-crystallization [36]. By combining these components, copolymers can exhibit properties that cannot be attained by single polymers, thereby leading to the creation of novel materials. Therefore, it is necessary to comprehend the structure–property correlation of melt-spun PB-1 filaments, as well as the distinctions between homopolymer and copolymer filaments.

In this study, PB-1 filaments of its homopolymer and copolymer with 4 mol% ethylene co-units were produced via extrusion melt spinning. The variances in microstructure, crystallization–melting behavior, and mechanical properties between homopolymer and copolymer filaments were analyzed using SEM, SAXS/WAXD, DSC, and tensile tests. The results show the clear structure–property correlation of melt-spun PB-1 filaments and are of guiding significance in developing a high-performance fiber with exceptional creep resistance.

## 2. Experimental Section

### 2.1. Materials

The samples used in this study are two kinds of polybutene-1 commercial materials, *Toppyl* PB8220M and *Toppyl* PB0110M, supplied by LyondellBasell, Rotterdam, The Netherlands. The grade PB8220M is a random copolymer of butene-1 with 4.3 mol% ethylene co-unit, of which the weight-averaged molecular weight (*M*_w_) is 414 kg/mol and the polydispersity index (PDI, *M*_w_/*M*_n_) is 2.7. The polybutene-1 grade PB0110M is a semicrystalline homopolymer, whose *M*_w_ is 742 kg/mol and PDI is 3.3. The melt flow rate of PB8220M and PB0110M is 2.5 g/10 min and 0.4 g/10 min (190 °C/2.16 kg), respectively.

### 2.2. Preparation of Filaments and Films

Spinning process of PB-1 samples is carried out with a melt-spinning system that consists of a single screw extruder and a spin pack, as shown in Figure 1a. PB-1 pellets are aged at 80 °C for 10 h in vacuum drying oven for water removal before spinning. The dried PB-1 samples are melted in the extruder and extruded into a spin pack where the molten polymer is metered and pressurized. The PB-1 filaments are ejected out of the spinneret and are drawn by a winding roll. The barrel temperature during extrusion of PB-1 melts is set at 110–260 °C. The take-up speed at winding roll is 7 and 10 m/min for PB0110M and PB8220M filaments, respectively.

The schematic diagram of hot pressing to PB-1 films is shown in Figure 1b. The PB-1 pellets are carefully positioned within a square-shaped mold with a thickness of 0.4 mm. Subsequently, the melting process of PB-1 samples are undertaken at a temperature of 180 °C to transform into homogeneous melt. After 10 min melting at 180 °C, the samples are taken out to cool naturally followed by aging at room temperature for one week. These films were cut into dumbbell-shaped samples for the stretching experiments.

### 2.3. Characterizations

The surface morphology of PB-1 filaments are observed by scanning electron microscopy (SEM, Hitachi S-3400, Tokyo, Japan) with an accelerating rate of 10 kV. The filament samples are sputter coated with Au to form a thin layer before SEM observation. The diameter of PB-1 monofilaments are measured by field emission scanning electron microscopy (FE-SEM, Thermo Fisher Apreo 2S HiVac, Brno, Czech Republic) with an accelerating rate of 10 kV. Each sample is measured 20 times and the average value is calculated.

Thermal analysis of PB-1 filaments and PB-1 films are conducted with a differential scanning calorimeter (DSC3, Mettler Toledo Instruments, Greifensee, Switzerland) under N_2_ atmosphere at a flow rate of 50 mL/min. The heating rate is 10 K/min. Each sample is measured 5 times and the average values of melting point and crystallinity are calculated, which are presented in Table 1.

Time-resolved SAXS/WAXD tests are carried out using Xeuss 3.0 SAXS/WAXS system, France, with the aid of a hybrid pixel detector (Eiger2R 1M, Dectris, Baden, Switzerland). It has a wavelength of 1.54 Å. SAXS and WAXD have sample-to-detector distances of 1800 mm and 63 mm, respectively. For all measurements, exposure times are 120 s for WAXD and 600 s for SAXS. XSACT (v2.7.33) software is used to convert the 2D pattern into a 1D azimuthal angle integration pattern, and the Fit2D (v12.077) software is employed to integrate the 2D–WAXD images into 1D curves. The full width at half-maxima (*FWHM*) of the 1D azimuthal angle integration curves are fitted by Gauss functions, and the degree of orientation (*π*) is derived by the following equation [37]:(1)$$\pi =\frac{180-FWHM}{180}$$

The degree of orientation (*π*) formula calculates the orientation using the *FWHM* of the peaks in the plot of the intensity versus azimuthal angle at the relevant 2*θ* value.

The mechanical properties of PB-1 filaments and PB-1 films are measured by tensile stretch on a material testing machine (ZwickRoell, Ulm, Germany) using a 20 N and 10 kN Xforce load cell at room temperature, respectively. The initial distance between the clamps on the sample is 20 mm. The PB-1 monofilament’ tensile stretch tests are carried out in accordance with the standard ASTM D 3822/D 3822 M [38] at a drawing rate of 5 mm/min. The PB-1 films’ tensile stretch tests are carried out in accordance with the standard ASTM D 882 [39] at a drawing rate of 20 mm/min. Tensile tests on PB-1 monofilament and PB-1 film are performed 20 times and 5 times, respectively.

## 3. Results and Discussion

### 3.1. Surface Morphology Characterization

The surface morphology of PB-1 filaments was observed by SEM, as shown in Figure 2. Figure 2a,b presents the micrographs of PB0110M filaments, showing that the filaments had a uniform diameter and rough surface. From Figure 2c,d, the surface microstructure of PB8220M filaments can be observed, of which the diameters were uniform while the surface was less rough than PB0110M filaments. The diameter of PB8220M filaments was about 120 μm (123.18 ± 3.71 μm), which is half of the diameters of PB0110M filaments (269.38 ± 14.25 μm). A rough surface similar to the PB8220M filaments was also observed on the carbon fiber surface. The unbalanced contraction between the core and shell of filaments during cooling and shear was considered to trigger the wrinkle of surface [40]. It has been reported that the surface roughness has little effect on the strength and modulus of carbon fiber as the linear density and diameter are much the same [41], which could be excluded as one of the potential factors affecting the mechanical properties in our study. Appendix A shows the SEM micrographs of the cross-sections of PB-1 filaments, and the FE-SEM micrographs of the surface of PB-1 films are shown in Appendix A. It is observed in Appendix A that the cross-sections of both filaments are circular and no obvious defects appear. It is seen in Appendix A that the surface of the PB8220M film presents many small sheet-like structures attached, while the surface of the PB0110M film is smooth and flat. It may be because the ethylene comonomer does not participate in the crystallization of the polybutene-1 chain segments and is excluded into the amorphous region. The differences shown by the two films were not significantly observed in the filaments. Therefore, it is reasonable to assume that the mechanical properties are minimally affected by the surface roughness of PB-1 filaments.

### 3.2. Thermal Analysis

The DSC melting curves of PB-1 filaments and PB-1 films were recorded with a heating rate of 10 K/min, and the heating curves of PB-1 after non-isothermal crystallization from homogeneous melt followed by aging at room temperature for one week were measured for comparison, as presented in Figure 3. All the samples are in form I state, as proved by the WAXD patterns in Figure 4. PB0110M samples had a higher melting temperature (*T*_m_) and crystallinity than the PB8220M samples. The flexible linear polybutene-1 chains have a strong tendency to crystallize at the temperature below melting point. In the random copolymer with ethylene co-unit, the successive polybutene-1 chains were disrupted by ethylene co-units such that the length of crystallizable polybutene-1 chain segments decreased, leading to a decrease in melting point and crystallinity [42]. Table 1 shows the statistical analysis results for the melting temperature and crystallinity obtained from the DSC curves in Figure 3. The crystallinity values were calculated based on *Φ*_w_ = Δ*H*/Δ*H*_ideal_, where Δ*H*_ideal_ is 116 J/g for form I [43]. According to Table 1, in comparison to PB0110M films and PB8220M films, the crystallinity of their filaments increased by around 9% and 7%, respectively. On the one hand, the increase in the crystallinity of PB-1 filaments is attributed to the application of shear forces to the polymer melt during the melt-spinning process, which can induce chain alignment and promote crystallization. The stretching and twisting of the polymer chains under these forces encourage the formation of crystalline regions [44,45]. On the other hand, the presence of ethylene co-units enhances the chain’s flexibility because polybutene-1 has ethyl side groups in each repetitive unit but polyethylene has no side group, which facilitates the ordering procedure and also increases crystallinity after the spinning process. When comparing PB0110M filaments with PB8220M filaments, it is evident that the presence of the ethylene co-units has a more significant impact on shortening the crystallizable segment length than on enhancing flexibility and ordering processes, resulting in a lower crystallinity in copolymer samples. The melting peak temperature of filaments in both PB0110M and PB8220M were slightly deviated from that of the films, because crystallization of filaments samples is expected to occur at lower temperature since filament-form melts cool faster after being ejected out of the spinneret directly into room temperature than homogeneous melt cooling at a constant rate of 10 K/min.

### 3.3. Microstructure Characterization

SAXS and WAXD analysis were used to measure the crystalline structure and orientation of polymer filament samples. Figure 4 shows the 2D–WAXD and 2D–SAXS patterns of PB-1 films and PB-1 filaments. The scattering vectors along and perpendicular to the filament axis direction are defined as *q*_1_ and *q*_2_, where 1 and 2 stand for the equator and meridian directions of scattering pattern, respectively (*q* = 4*π sin θ*/*λ*, where 2*θ* is the scattering angle and *λ* is the X-ray wavelength) [46]. One can conclude from the WAXD pattern that the crystalline region of PB0110M filaments was approximately isotropic and did not possess the distinct orientation, which is consistent with the isotropic rings shown by the PB-1 films. However, the scattering intensity of PB8220M filaments was found to slightly converge on the meridian, which indicates that crystalline domains are oriented along the *q*_1_ direction. In the SAXS analysis of PB-1 films (Figure 4c,d), we observed randomly arranged alternating layers of lamellar crystals and amorphous regions. These layers displayed isotropic signals. The structure information of PB-1 filaments can be observed at a small angular scale from the SAXS of Figure 4g,h. The approximate isotropic SAXS pattern of PB0110M filaments at room temperature is shown in Figure 4g. It is a typical pattern of the low oriented lamellar structure, namely, layers of lamellar crystals and amorphous areas alternating and oriented along the stretching direction [46]. The SAXS pattern of the PB8220M filaments shown in Figure 4h is anisotropic, characterized by a streak pattern in the meridian direction and a two-point pattern along the equator direction. The presence of the two-point pattern indicates that the lamellae are stacked perpendicular to the filament axis, while the streak pattern can be attributed to the presence of extended-chain crystals, also known as shish. This anisotropic SAXS pattern provides valuable information about the arrangement and orientation of different crystalline structures within the PB8220M filaments [47]. Therefore, the PB8220M filaments appear with the typical shish-kebab structures, with fibrous crystals and alternating crystal lamellae and amorphous regions.

The top part of Figure 5 and Figure 6a exhibit the 1D–WAXD curves of PB0110M filaments (black solid curve), PB8220M filaments (red solid curve), PB0110M films (black dashed curve), and PB8220M films (red dashed curve), respectively, which were calculated by Fit2D (v12.077). The primary diffraction peaks detected at 2*θ* of about 10.0°, 17.5°, and 20.4° correspond to crystallographic planes of (110), (300), and (220 + 211) of form I, respectively [48]. Furthermore, the (200), (220), and (213) lattice planes of form II can be assigned a sequence of reflections at 11.9°, 16.9°, and 18.5° [12], respectively. For PB-1 filaments and PB-1 films, the crystal phases are mainly in the stable form I after fibrillation and aging at room temperature. Nevertheless, diffraction peaks of (200)_II_ and (213)_II_ in PB0110M samples are also observed while PB8220M samples show pure form I diffraction peaks. Qiao [49] introduced that the degree of transition reaches a plateau value in the late stage of transition; low molecular weight always contributes to a greater degree of transition. A possible explanation is that entanglements and intercrystalline links in high-molecular-weight systems play a role in stabilizing the metastable form II by preventing chain translational motion in the crystalline phase and delaying the relaxation of amorphous chains during phase transition [50]. On the other hand, the presence of ethylene co-units accelerates the transition of form II into form I at room temperature. There is a suggestion that a concentration of approximately 6 mol% of ethylene co-units promotes the direct crystallization of stable form I from the melt [51]. The thermodynamically stable form I is expected to result in good mechanical properties for PB8220M filaments.

To evaluate the mechanical properties of PB-1 filaments, the degree of orientation is considered to be an important factor. The attainment of optimal mechanical strength in filaments can be realized through the formation of totally extended, ultra-long molecular chains. The extent to which the actual conformation of molecular chains in PB-1 filaments deviates from the ideal extended chain model can be accurately assessed by quantifying the orientation parameter of said filaments [52,53]. WAXD analyses were conducted to investigate the degree of orientation of the small crystals within filaments, while SAXS was employed to accurately quantify the orientation information of large microfibril structures. The 1D azimuthal integration curves are used to calculate the degree of orientation, as illustrated at the bottom of Figure 5 and in the middle of Figure 7. The structural parameters are listed in Table 2. We surprisingly found that the PB8220M filaments exhibited a greater degree of orientation in comparison with the PB0110M filaments. PB0110M samples have higher molecular weight and possess longer chains, leading to high content of inter-and intramolecular entanglements. Ethylene co-units are a linear structure without the ethyl side group. It is easier to disentangle the copolymer and to make the molecular chain more flexible when it is in a molten state [54]. During the melt-spinning process, the PB8220M sample has better fluidity and melting processability due to its low melting point and high melt flow rate. Meanwhile, because of a low winding speed, the PB0110M filaments was formed by gravity and incomplete drawing, resulting in a low degree of orientation. Besides the higher content of form I, a highly oriented fibrillar structure of the initial PB8220M filaments is one of the reasons for their high mechanical properties [55].

Figure 6b and the top part of Figure 7 show the Lorentz-corrected 1D-scattering intensity distributions of the PB-1 films and PB-1 filaments, independently. The scattering peaks indicate a regularly stacked lamellar crystalline structure within the polymer samples. The thicknesses of crystalline lamellae (*d*_c_), amorphous layers (*d*_a_), and long spacing (*d*_ac_) of PB-1 in form I crystalline structure were determined using the correlation function approach [49], shown in Figure 6c and Figure 7 (bottom); the values obtained are listed in Table 2. Because the crystallinity of PB0110M samples was larger than 50%, the smaller value obtained from the intersection point of the correlation function was assigned to *d*_a_, and *d*_c_ was obtained by the equation *d*_c_ = *d*_ac_ − *d*_a_. For PB8220M samples with crystallinity lower than 50%, the value at intersection point corresponds to *d*_c_, and *d*_a_ = *d*_ac_ − *d*_c_. The results indicate that PB0110M samples have larger lamellar thickness and long spacing, which is in accordance with the higher crystallinity results in DSC analysis. Random incorporation of ethylene co-units erodes the integrity of polybutene-1 chains and the length of the crystallizable chain decreases, resulting in a thinner lamellar thickness. Moreover, ethylene co-units were excluded into the amorphous phase during the crystallization of butene-1 chains in the copolymer, which also contributes to a larger thickness of the amorphous phase in PB8220M.

### 3.4. Mechanical Properties

Figure 8 and Figure 9 show the stress–strain behavior for PB-1 filaments and PB-1 films, respectively. The mechanical properties of PB0110M filaments and PB8220M filaments are compared in Figure 8. The tensile strength of the PB8220M filament is four times that of the PB0110M filament, and its Young’s modulus is 1.5 times that of the PB0110M filament, which could be attributed to the high content of form I and the degree of orientation in PB8220M filament even though it has a lower crystallinity and lamellar thickness. For polymer films prepared by hot-press, the homopolymer PB0110M exhibits higher Young’s modulus and yield stress, and slightly higher tensile strength, than the copolymer PB8220M, which is expected to be related to the crystallinity and lamellar thickness. PB8220M filaments exhibited higher tensile strength, yield stress, and Young’s modulus than PB0110M filaments while the PB0110M sample showed better mechanical properties in the film state. Due to the good flexibility of ethylene co-units molecular chains in the amorphous phase, the conformational changes or orientation of the molecular chains occurred under tensile stress [56]. The PB8220M filaments with a longer shish length were supposed to show higher tensile strength. However, the elongation-at-break of the PB0110M filaments versus PB8220M filaments was not significantly different. The elongation-at-break of PB8220M films was slightly greater than that of PB0110M films. Different filament diameters may also account for differences in mechanical properties between homopolymer and copolymer filaments.

## 4. Conclusions

In this work, the filaments of polybutene-1 and its random copolymer with 4 mol% ethylene co-units were produced via extrusion melt spinning. The variances in microstructure, crystallization–melting behavior, and mechanical properties between homopolymer and copolymer filaments were analyzed using SEM, SAXS/WAXD, DSC, and tensile tests. The crystallization of form II and subsequent phase transition into form I finished after the melt-spinning process in copolymer sample, while a small amount of form II remained in homopolymer filament. The diameters of both filaments were uniform, and the surface of copolymer filaments was less rough than homopolymer filaments. The copolymer filaments exhibit a tensile strength approximately four times greater than that of the homopolymer and Young’s modulus approximately 1.5 times higher. Despite the potential disruption of PB-1 chains due to the random incorporation of ethylene copolymerized units, which resulted in shortened crystallizable chains and reduced lamellae thickness; the presence of these units also enhances chain flexibility. This flexibility proves advantageous for the formation of intercrystalline chains during spinning, leading to copolymer filaments with increased orientation and longer fiber crystals. Consequently, the melt-spun copolymer filaments demonstrate superior tensile properties. This part of the work has certain implications for expanding the application field of PB-1, such as the development of a high-performance polymer fiber with outstanding creep resistance.

## Figures and Tables

**Figure 1 polymers-15-03729-f001:**
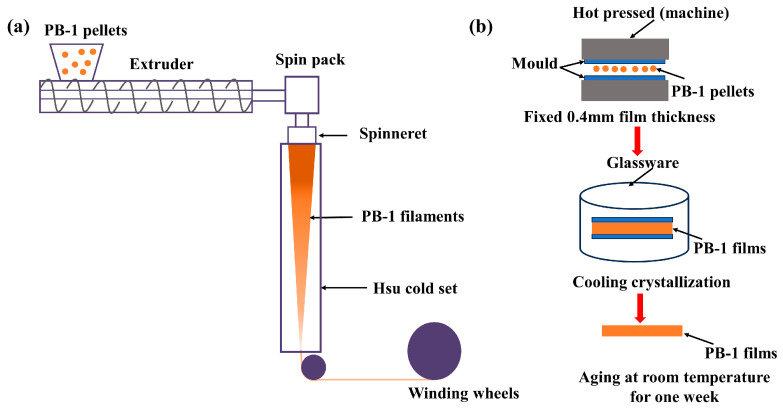
Schematic diagram of melt-spinning system to fabricate PB-1 filaments (**a**) and hot pressing to PB-1 films (**b**).

**Figure 2 polymers-15-03729-f002:**
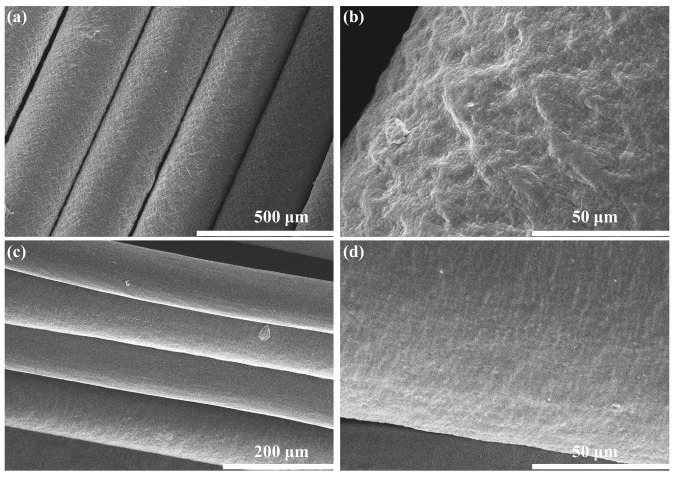
SEM micrographs of PB0110M filaments (**a**,**b**) and PB8220M filaments (**c**,**d**).

**Figure 3 polymers-15-03729-f003:**
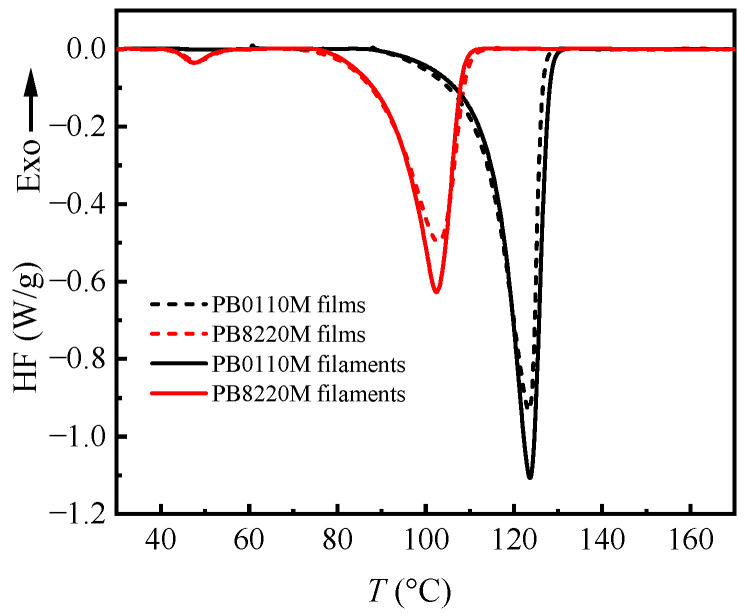
DSC heating curves of PB-1 films and PB-1 filaments at a rate of 10 K/min.

**Figure 4 polymers-15-03729-f004:**
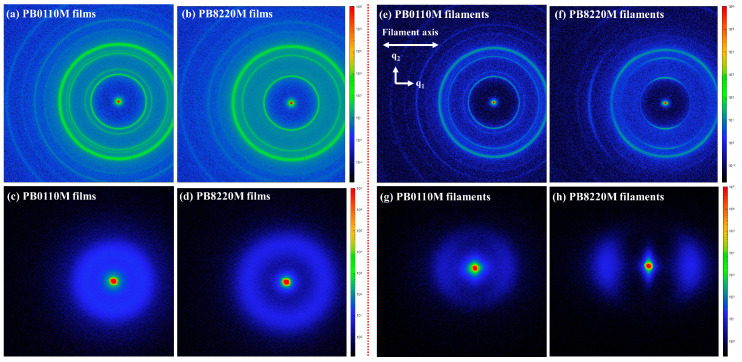
2D–WAXD (**a**,**b**,**e**,**f**) and 2D–SAXS (**c**,**d**,**g**,**h**) patterns of PB-1 films and PB-1 filaments: (**a**,**c**) PB0110M films; (**b**,**d**) PB8220M films; (**e**,**g**) PB0110M filaments; (**f**,**h**) PB8220M filaments.

**Figure 5 polymers-15-03729-f005:**
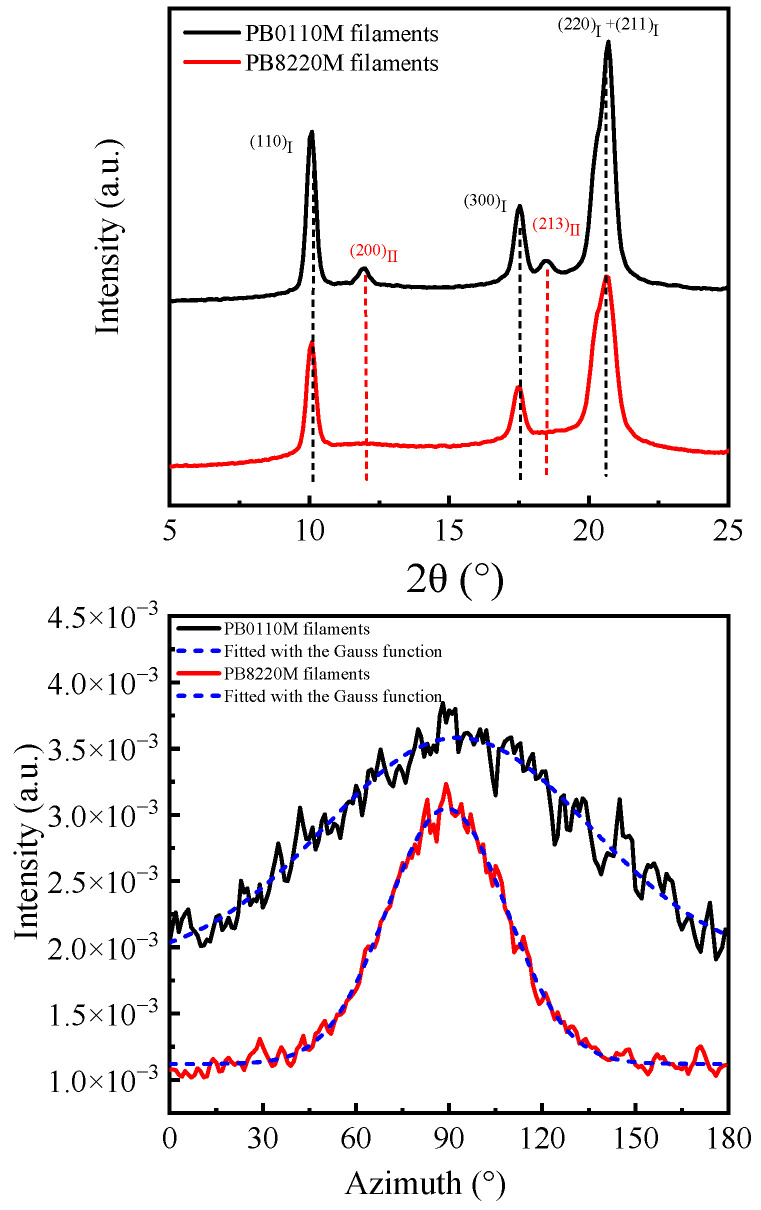
1D–WAXD curves of PB-1 filaments (*λ* = 0.154 nm) (**top**); 1D–WAXD azimuthal integration signals (**bottom**).

**Figure 6 polymers-15-03729-f006:**
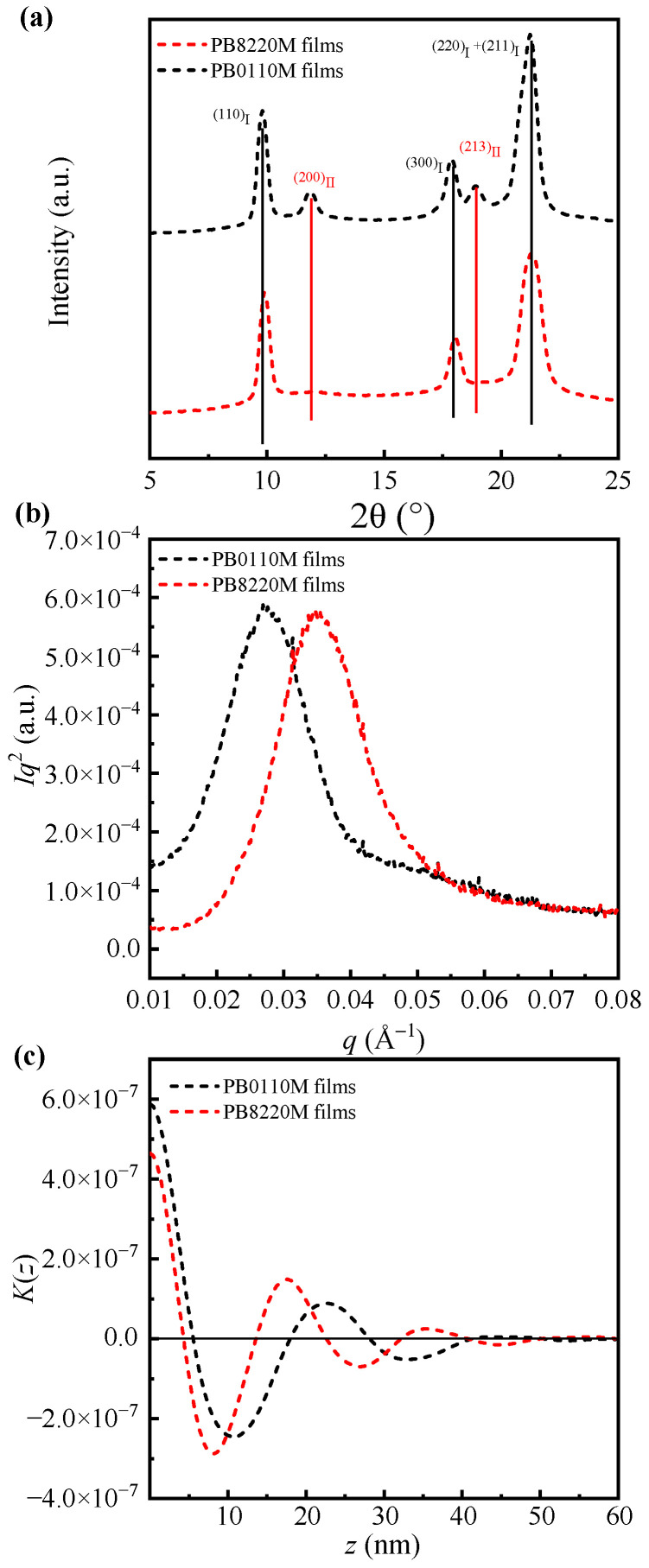
1D–WAXD curves of PB-1 films (λ = 0.154 nm) (**a**). 1D–SAXS intensity distribution profiles (**b**). 1D–electron density correlation function curves (**c**).

**Figure 7 polymers-15-03729-f007:**
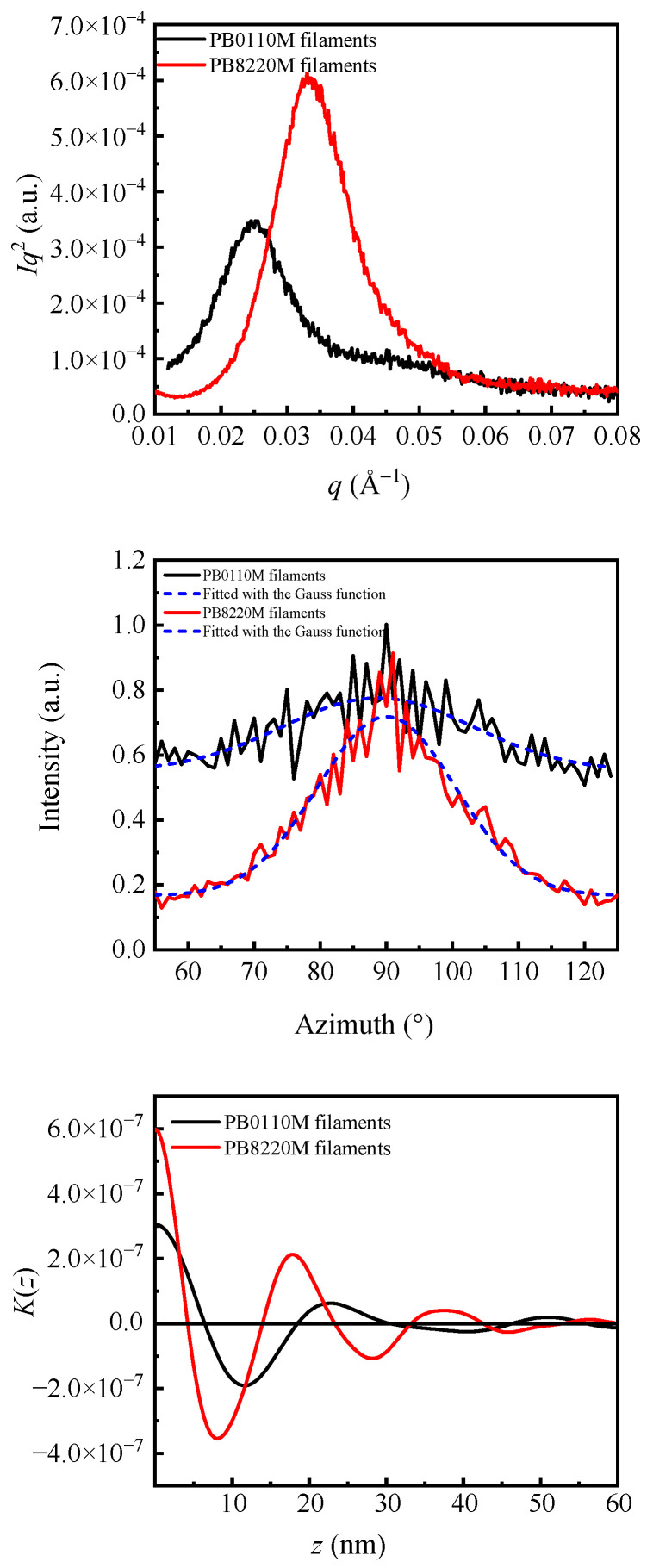
1D–SAXS intensity distribution profiles (**top**). 1D–SAXS azimuthal integration signals (**middle**). 1D–electron density correlation function curves (**bottom**).

**Figure 8 polymers-15-03729-f008:**
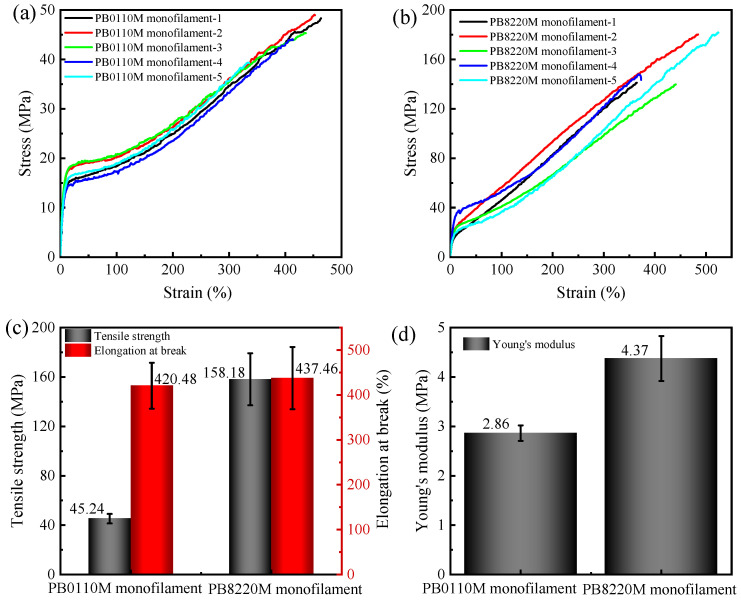
Mechanical properties of PB-1 filaments at room temperature: (**a**) stress–strain curves of PB0110M monofilament; (**b**) stress–strain curves of PB8220M monofilament; (**c**) tensile strength and elongation at break; (**d**) Young’s modulus. Tensile tests on PB-1 monofilament were performed 20 times; the stress–strain curves are presented with 5 pieces per sample.

**Figure 9 polymers-15-03729-f009:**
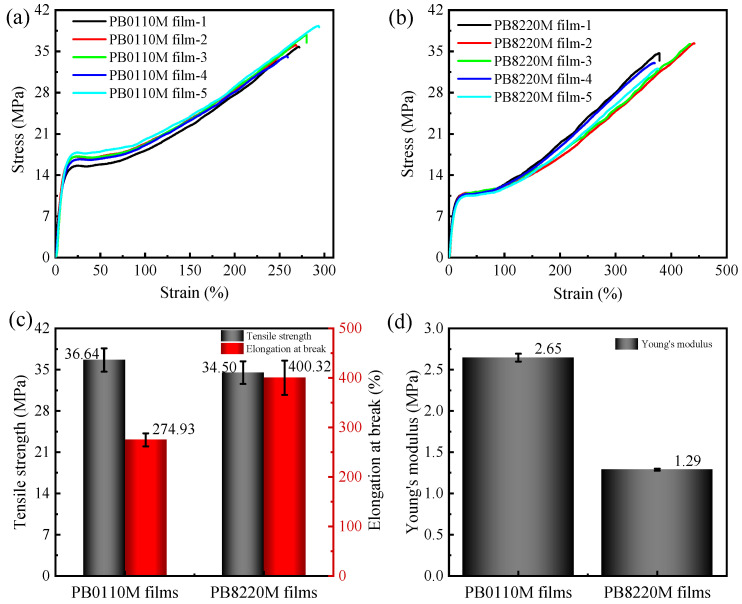
Mechanical properties of PB-1 films at room temperature: (**a**) stress–strain curves of PB0110M film; (**b**) stress–strain curves of PB8220M film; (**c**) tensile strength and elongation at break; (**d**) Young’s modulus. Tensile tests on PB-1 films were performed 5 times; the stress–strain curves are presented with 5 pieces per sample.

**Table 1 polymers-15-03729-t001:** Crystallinity and melting temperature obtained from DSC curves.

Samples *	*T*_m_/°C	Crystallinity/%
PB0110M films	122.95 ± 0.86	52.19 ± 1.43
PB8220M films	101.31 ± 1.39	36.46 ± 1.05
PB0110M filaments	123.67 ± 0.09	56.91 ± 1.63
PB8220M filaments	102.38 ± 0.58	39.15 ± 0.45

* Each sample is measured 5 times and the average values are calculated and presented in Table 1. The differences in the results between the four materials were caused by the different processing conditions (hot pressing and melt spinning) of the samples.

**Table 2 polymers-15-03729-t002:** Structural parameters analyzed from 2D–WAXD and SAXS patterns of PB-1 films and PB-1 filaments.

Samples	WAXD Orientation along (110)		SAXS Orientation along *q*_2_		Long Spacing	Lamellar Thickness	Amorphous Layers
	*FWHM* (°)	*π* (%)	*FWHM* (°)	*π* (%)	*d*_ac_ (nm)	*d*_c_ (nm)	*d*_a_ (nm)
PB0110M films	-	-	-	-	22.58	15.37	7.21
PB8220M films	-	-	-	-	17.61	6.28	11.33
PB0110M filaments	100.47	44.18	34.05	81.08	22.68	13.37	9.31
PB8220M filaments	45.38	74.79	24.57	86.35	17.82	6.20	11.62

## Data Availability

Data reported in this paper are available from the authors upon reasonable request.

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
