# Peer review of "Microstructure and Tensile Properties of Melt-Spun Filaments of Polybutene-1 and Butene-1/Ethylene Copolymer"

_polymers, 2023, doi:10.3390/polym15183729_

Round 1

Reviewer 1 Report

In this article, the authors investigated the properties of Polybutene-1 with form I crystals and its interaction with a 4 mol% ethylene copolymer. While the study offers valuable insights into the material's microstructural and mechanical attributes through methods like SEM, SAXS/WAXD, and DSC, it requires major revisions to meet the standards for publication. The differentiation between copolymer fibers and homopolymer, especially in the context of their mechanical properties, needs clearer exposition. The role of ethylene in influencing chain flexibility and the development of intercrystalline links during spinning, while interesting, requires more robust evidence and elaboration. Only with comprehensive modifications can this article be deemed publishable.

 Abstract

- The abstract mentions that the "homopolymer showed better mechanical properties in film state," but does not compare these properties in the copolymer or explain why the homopolymer’s properties differ between the film state and fiber state.

- While the abstract ends with a statement about the ethylene's contribution, it could benefit from a more explicit concluding sentence summarizing the study's broader implications or significance.

 Introduction

- The introduction seems to move between topics a bit abruptly. Establishing a clearer structure will help readers follow along and understand the progression of information.

- There are some repeated themes or points. For instance, the value of PB-1 (its excellent properties) is reiterated multiple times. Reducing repetition would help shorten the introduction without losing essential information.

- The authors stated, "The spinning process entails applying a drawing force to the polybutene-1 melt, followed by crystallization through stretching. During this process, polymer molecules and crystals align parallel to the fiber axis, which imparts unique properties to the resulting filaments." However, this description specifically pertains to melt-spinning and not spinning as a whole. It would be more accurate for the authors to specify "melt-spinning" instead of using the broader term "spinning."

- While the last paragraph seems to detail the current study's objective, this could be made clearer earlier in the introduction. A strong thesis statement early on can give readers a clear idea of what to expect.

- The introduction could benefit from a more explicit concluding statement that ties together the context provided with the aims of the current study. While this is somewhat provided, making it distinct and punchy will create a stronger impression.

Results

- Observations like the "unbalanced contraction between core and shell of fibers" could benefit from further clarification or background to be more accessible to readers unfamiliar with the topic.

- The section includes references [31] and [32], but it's not entirely clear how these references relate to the work being presented. For instance, the statement about carbon fiber might leave readers wondering how directly comparable the behavior of PB-1 fibers is to carbon fibers.

- The transition from the description of PB-1 fibers to the mention of carbon fiber is somewhat abrupt. This shift could be smoothed with a sentence or two that bridges these ideas.

- It is better that the authors provide the statistical analysis results for thermal characterization by DSC.

- "The authors noted that introducing additional constituents into the copolymer disrupts its regularity and shortens the crystallization chain segments, leading to a decrease in both melting point and crystallinity. However, in a separate statement, they highlighted that the presence of ethylene co-units enhances the chain's flexibility. This improved flexibility aids the ordering process, resulting in increased crystallinity after the spinning process. These explanations appear to be misaligned."

- The authors should provide a detailed comparison of the crystallinity observed in two different fibers.

- The authors observed that PB0110M fibers possess longer molecular chains, which are prone to tangling, hindering their optimal alignment. In contrast, PB0110 fibers exhibit greater crystallinity. This presents a seeming inconsistency, as typically, proper chain orientation is a precursor to the formation of crystallinity.

- In the mechanical properties section, the authors highlighted that “the tensile strength of the PB8220M fiber is quadruple that of the PB0110M fiber, and its Young's modulus is 1.5 times greater. This disparity is attributed to the high content of form I and degree of orientation in the PB8220M fiber” while typically, an adequate chain orientation acts as a foundation for the development of crystallinity. It's essential for the authors to specify where they observe a higher degree of orientation. Are they referring to the amorphous phase? Moreover, it's commonly understood that the overall crystallinity usually plays a pivotal role in determining the mechanical properties of fibers. Given this, one would expect the PB8220M fibers, having greater crystallinity, to exhibit superior mechanical attributes.

 Moderate editing of English language required

Reviewer 2 Report

·       Include PB-1 foils, SEM and DSC in the key words of the paper. Include the word filaments or monofilaments instead of fiber in the title and text of the paper.

·      Introduction: It is necessary to highlight the aim of the manuscript more clearly. Text in page 2 lines 87 - 95 move to the Conclusion. Also, in the Introduction, give a more comprehensive description of the topics and conducted research directly related to the topic of the paper. 

·    Materials and Methods: PB-1 foils production needs to be explained in more detail. It is necessary to show the schematic diagram of PB-1 foils production. How is the diameter of filaments and thickens of the foils determined?  For SEM microscopy magnification level is missing. For standardized test methods, it is necessary to describe the implementation of the test in details and the presentation of the results. E.g. for tensile tests number of measurements, distance between the clamps is missing.

·   Results and discussion: 3.1. Describe the measurement of filaments diameter and indicate the number of measurements taken.  Line 152 – for this conclusion linear density and CV for diameter value of filaments should be presented. The shape of filaments cross section should be mentioned.

·      In Figures 7 and 8 (c, d) add numerical values. Write that the average values were obtained by testing on (?) samples. In figures 7 and 8 (a, b) single measurement are presented? Page 9, line 277 - lamellar thickness and the crystallinity of foils are not measured.

Generally, should be explained – why the measurements within (3.1., 3.3.) were performed on filaments only, (3.2) on pellets and filaments and (3.4.) on filament and foils.

·     Conclusion: It is necessary to specify the possible application of the obtained materials.

Reviewer 3 Report

The manuscript studies the differences in the micro-morphology, in crystallinity, and in tensile properties between the homopolymer and copolymer of polybutene-1 in the shape of fibers and films. The study is not cutting edge, since many previous similar analyses have been published, as can be noted by the rich bibliography analyzed. Nevertheless, the referee cannot find in the scientific literature the particular combination of aspects addressed in this manuscript. Moreover, the manuscript is well-written and well-organized. Therefore, in the referee’s opinion, it is acceptable for publication.

In the referee's opinion, the results of the tensile testing currently represented in Figures 7 (c-d) and 8 (c-d) should be explicitly numerically expressed in order to provide a benchmark to the readers.

Round 2

Reviewer 1 Report

- The authors should include standard deviations for the data presented in Table 1 and explicitly state whether the differences observed in Table 1 are statistically significant.

- The " increase in crystallinity" following the spinning process is anticipated in both polymers and copolymers as evidenced by the data presented in Table 1. This effect is attributed to the applied shear forces to the polymer melt during the melt-spinning process which can induce chain alignment and promote crystallization. The stretching and twisting of the polymer chains under these forces encourage the formation of crystalline regions. As a result, the increased crystallinity observed in copolymer fibers, as compared to copolymer films, cannot be solely attributed to the enhanced flexibility and polymer chain order resulting from the presence of ethylene co-units.

- It is nessessary to mentioned in the manuscript that the impact of ethylene co-units on shortening the crystallizable segment length was more pronounced than their influence on enhancing flexibility and the ordering process. This distinction becomes evident when comparing the lower crystallinity observed in copolymer fibers to that of polymer fibers.

Minor editing of English language required

Reviewer 2 Report

 Accept in present form

Author Response

Dear Reviewer:

Thanks very much for your kind work and consideration of the publication of our paper. On behalf of my co-authors, we would like to express our great appreciation to the reviewer.
Thank you and best regards.
Yours sincerely,

Dr. Yongna Qiao

Institute of Low-Dimensional Materials Genome Initiative, College of Chemistry and Environmental Engineering of Shenzhen University

Address: 3688 Nanhai Ave, Shenzhen, Guangdong, P.R.China, 518060

Tel: 86-755-26535427

Fax:86-755-26536141

Email: yongna_qiao@sina.com

Round 3

Reviewer 1 Report

The manuscript in the current version is acceptable for publication.

Minor edit is required